# Golgi associated RAB2 interactor protein family contributes to murine male fertility to various extents by assuring correct morphogenesis of sperm heads

Haoting Wang[1,2], Rie Iida-Norita[2], Daisuke Mashiko[2], Anh Hoang Pham[1,2], Haruhiko Miyata[2]*, Masahito Ikawa[1,2,3,4,5]*

1 Graduate School of Pharmaceutical Sciences, Osaka University, Suita, Osaka, Japan, 2 Research Institute for Microbial Diseases, Osaka University, Suita, Osaka, Japan, 3 Graduate School of Medicine, Osaka University, Suita, Osaka, Japan, 4 The Institute of Medical Science, The University of Tokyo, Minato-ku, Tokyo, Japan, 5 Center for Infectious Disease Education and Research, Osaka University, Suita, Osaka, Japan

* hmiya003@biken.osaka-u.ac.jp (HM); ikawa@biken.osaka-u.ac.jp (MI)

⊙ OPEN ACCESS

**Data Availability Statement:** All relevant data can be found within the article and its supplementary information files.

## Abstract

Sperm heads contain not only the nucleus but also the acrosome which is a distinctive cap-like structure located anterior to the nucleus and is derived from the Golgi apparatus. The Golgi Associated RAB2 Interactors (GARINs; also known as FAM71) protein family shows predominant expression in the testis and all possess a RAB2-binding domain which confers binding affinity to RAB2, a small GTPase that is responsible for membrane transport and vesicle trafficking. Our previous study showed that GARIN1A and GARIN1B are important for acrosome biogenesis and that GARIN1B is indispensable for male fertility in mice. Here, we generated KO mice of other *Garins*, namely *Garin2*, *Garin3*, *Garin4*, *Garin5a*, and *Garin5b* (*Garin2-5b*). Using computer-assisted morphological analysis, we found that the loss of each *Garin2-5b* resulted in aberrant sperm head morphogenesis. While the fertilities of *Garin2*⁻/⁻ and *Garin4*⁻/⁻ males are normal, *Garin5a*⁻/⁻ and *Garin5b*⁻/⁻ males are subfertile, and *Garin3*⁻/⁻ males are infertile. Further analysis revealed that *Garin3*⁻/⁻ males exhibited abnormal acrosomal morphology, but not as severely as *Garin1b*⁻/⁻ males; instead, the amounts of membrane proteins, particularly ADAM family proteins, decreased in *Garin3* KO spermatozoa. Moreover, only *Garin4* KO mice exhibit vacuoles in the sperm head. These results indicate that GARINs assure correct head morphogenesis and some members of the GARIN family function distinctively in male fertility.

## Author summary

Approximately 5% of males exhibit infertility, which has become a significant problem worldwide. One major reason for male infertility is teratozoospermia, characterized by spermatozoa with abnormal morphology. However, the genetic cause of teratozoospermia is largely unknown. In our previous study, we found two proteins belonging to the Golgi

**Funding:** This research was supported by the Ministry of Education, Culture, Sports, Science and Technology (MEXT)/Japan Society for the Promotion of Science (JSPS) KAKENHI grants (https://www.jsps.go.jp/english/e-grants/index.html, JP22H03214, JP23K18328 to H.M. and JP19H05750, JP21H04753, JP21H05033 to M.I.); Takeda Science Foundation grant (https://www.takeda-sci.or.jp/) to H.M. and M.I.; JST FOREST (https://www.jst.go.jp/souhatsu/en/index.html, JPMJFR211F to H.M.); the Eunice Kennedy Shriver National Institute of Child Health and Human Development (https://www.nichd.nih.gov/, P01HD087157 and R01HD088412 to M.I.); and the Bill & Melinda Gates Foundation (https://www.gatesfoundation.org/, Grand Challenges Explorations grant INV-001902 to M.I.). The funders had no role in study design, data collection and analysis, decision to publish, or preparation of the manuscript.

**Competing interests:** The authors have declared that no competing interests exist.

associated RAB2 interactor (GARIN) protein family are responsible for sperm acrosome biogenesis and mutation of each one of them causes globozoospermia in mice, which is a subset of teratozoospermia defined as spermatozoa that show round sperm heads with abnormal acrosome morphology. In this study, we generated knockout mice of each gene belonging to the *Garin* family. By applying elliptic Fourier descriptors (EFDs) and principal component (PC) analysis on sperm heads, we demonstrated that GARINs are all responsible for correct sperm head morphogenesis. GARINs are all conserved in humans, and expanding the understanding of GARINs potentially contributes to the elucidation of human male infertility.

## Introduction

Spermatozoa are generated through a specialized biological process known as spermatogenesis. This intricate process occurs in seminiferous tubules of the testis, comprising a series of discrete steps. At the first step, spermatogonia undergo mitosis, transitioning into spermatocytes. Meiosis then occurs to transform diploid spermatocytes into haploid round spermatids. Spermiogenesis follows meiosis and is characterized by sixteen steps in mice. Spermiogenesis is a complicated process that involves nuclear condensation, morphogenesis of the sperm head, and formation of the sperm tail. These events collectively ensure the integrity of spermatozoa and secure the correct interaction of spermatozoa with oocytes.

The acrosome, a distinctive Golgi-derived structure situated at the anterior aspect of the sperm head, plays an important role in sperm-egg interaction [1–3]. The acrosome reaction, characterized as an exocytosis event releasing the enzymes stored within the acrosome, assists sperm penetration through the zona pellucida (ZP) of oocytes [1,2,4]. Simultaneously, the acrosome reaction assists the migration of receptors that facilitate sperm-egg fusion [5–7]. Spermiogenesis can be divided into 4 phases based on the morphology of the acrosome and nucleus: Golgi phase, cap phase, acrosome phase, and maturation phase [8]. Starting from the Golgi phase, the Golgi apparatus actively produces vesicles containing various glycoproteins, and then, multiple vesicles fuse to form a larger vesicle which is called an acrosomal vesicle near the nucleus [9]. In the cap phase, acrosomal vesicles spread over the surface and become flattened around the nucleus. After the cap phase, the skirt-like microtubule-based structure, the manchette is formed, which facilitates nuclear elongation during the acrosome phase [10]. The nucleus and acrosomal vesicle then further elongate to complete spermiogenesis in the maturation phase.

Teratozoospermia is a major clinical cause of male infertility, which is defined as over 85% of the spermatozoa carrying abnormal morphology in humans [11,12]. Moreover, globozoospermia, a subset of teratozoospermia, is specifically characterized by spermatozoa with abnormal acrosome morphology [11]. The genetic and molecular mechanisms underlying teratozoospermia are largely unknown, and the mouse knockout (KO) model is used as an outstanding animal model to elucidate the genetic cause. In our previous study, we identified two globozoospermia-related genes: Golgi Associated RAB2 Interactor 1A (*Garin1a*; *Fam71f2*) and *Garin1b* (*Fam71f1*) [13]. *Garin1a* and *Garin1b* are predominantly expressed in the testes and KO of *Garin1a* and *Garin1b* causes subfertility and infertility in males, respectively, due to abnormal acrosomal morphology [13]. Furthermore, the same study showed that GARIN1B binds to both RAB2A and RAB2B which are small GTPases with high amino acid sequence similarities and are involved in vesicle trafficking. In contrast, the functions of other GARIN paralogues, namely GARIN3, GARIN4, GARIN5A, and GARIN5B, in male fertility are not

clear although it has been shown recently that *Garin2* (*Fam71d*) is not essential for male fertility in mice [14].

In this study, we generated 5 single KO mice of *Garin2*, *Garin3*, *Garin4*, *Garin5a*, and *Garin5b* and aimed to elucidate the characteristics of GARINs. we demonstrated that GARIN2, GARIN3, GARIN4, GARIN5A, and GARIN5B are all important for sperm head morphogenesis, along with GARIN3 being vital for acrosome biogenesis and male fertility.

## Results

### *Garin2-5b* are expressed predominantly in mouse testes

In mice, there are 7 members of the GARIN protein family, and they all possess a RAB2-binding domain with high sequence similarity of amino acids (Figs 1A and S1A). Phylogenetic trees of mouse GARINs (S1B Fig) and human GARINs (S1C Fig) were generated by comparing amino acid sequences using Clustal Omega [15]. The mouse GARIN protein family is well conserved in humans while human GARIN6 is not conserved in mice. We also generated phylogenetic trees for coding sequences of mouse *Garins* (S1D Fig) and human *GARINs* (S1E Fig) and found that GARIN5A and GARIN5B were closely related in coding sequences but not in amino acid sequences, suggesting that there may be more non-synonymous substitutions in GARIN5A and/or GARIN5B. We performed RT-PCR using cDNAs obtained from adult mouse tissues to determine the expression pattern of *Garin2*, *Garin3*, *Garin4*, *Garin5a*, and *Garin5b* (hereafter referred to as *Garin2-5b*) and found that all *Garin2-5b* are predominantly expressed in the testis (Fig 1B). Because the first wave of spermatogenesis in postnatal testis allows the consecutive appearance of spermatogonia, spermatocytes, round spermatids, and elongated spermatids to be analyzed [16], we analyzed the expression of mouse *Garin2-5b* using cDNA obtained from 10–35 days postnatal testes. All *Garin2-5b* showed strong signals from 28 days postnatal (Fig 1C), suggesting that GARIN2-5B may function during spermiogenesis after meiosis. These results were confirmed by published single-cell RNA-sequencing (RNA-seq) datasets [17,18], as *Garin2-5b* are expressed predominantly in testes and spermatids. In contrast, *Rab2a* and *Rab2b* are expressed in several tissues and do not exhibit predominant expression in late spermiogenesis (S1F and S1G Fig).

To understand the interaction between GARIN2-5B and RAB2A/B, we expressed FLAG-tagged GARIN2-5B and PA-tagged RAB2A or RAB2B (Fig 1D) in HEK293T cells followed by co-immunoprecipitation (Co-IP) and Western blotting (WB). We confirmed that GARIN2-5B could interact with RAB2A and RAB2B (Fig 1E–1I). These results suggest that GARIN2-5B may function during spermiogenesis possibly with RAB2A and/or RAB2B.

### Knockouts of *Garin2-5b* impact male fertility with varying severity

To examine the function of GARIN2-5B *in vivo*, we generated 5 single KO mouse lines of each *Garin2-5b* with the CRISPR/CAS9 system. For *Garin2*, *Garin3*, *Garin4*, and *Garin5b*, we used pairs of guide RNA (gRNA) targeting almost the entire open reading frames (ORFs). The numbers of electroporated zygotes, transplanted embryos, pups born, and pups confirmed with a large deletion are summarized in S1 Table. The genotype of these mouse lines was confirmed by genomic PCR with indicated primers (S2A–S2C and S2E Fig). Further, by performing Sanger sequencing of the PCR products, we confirmed 24,044 bp, 2,755 bp, 1,663 bp, 13,436 bp deletions on *Garin2*, *Garin3*, *Garin4*, and *Garin5b*, respectively. For *Garin5a*, we used one gRNA targeting exon 2 of the ORF (S2D Fig) and obtained mice with a 7 bp deletion, which caused the frameshift mutation. This 7 bp deletion was confirmed by Sanger sequencing of the genomic PCR product. Genotyping of *Garin5a* KO mice was performed by digesting the

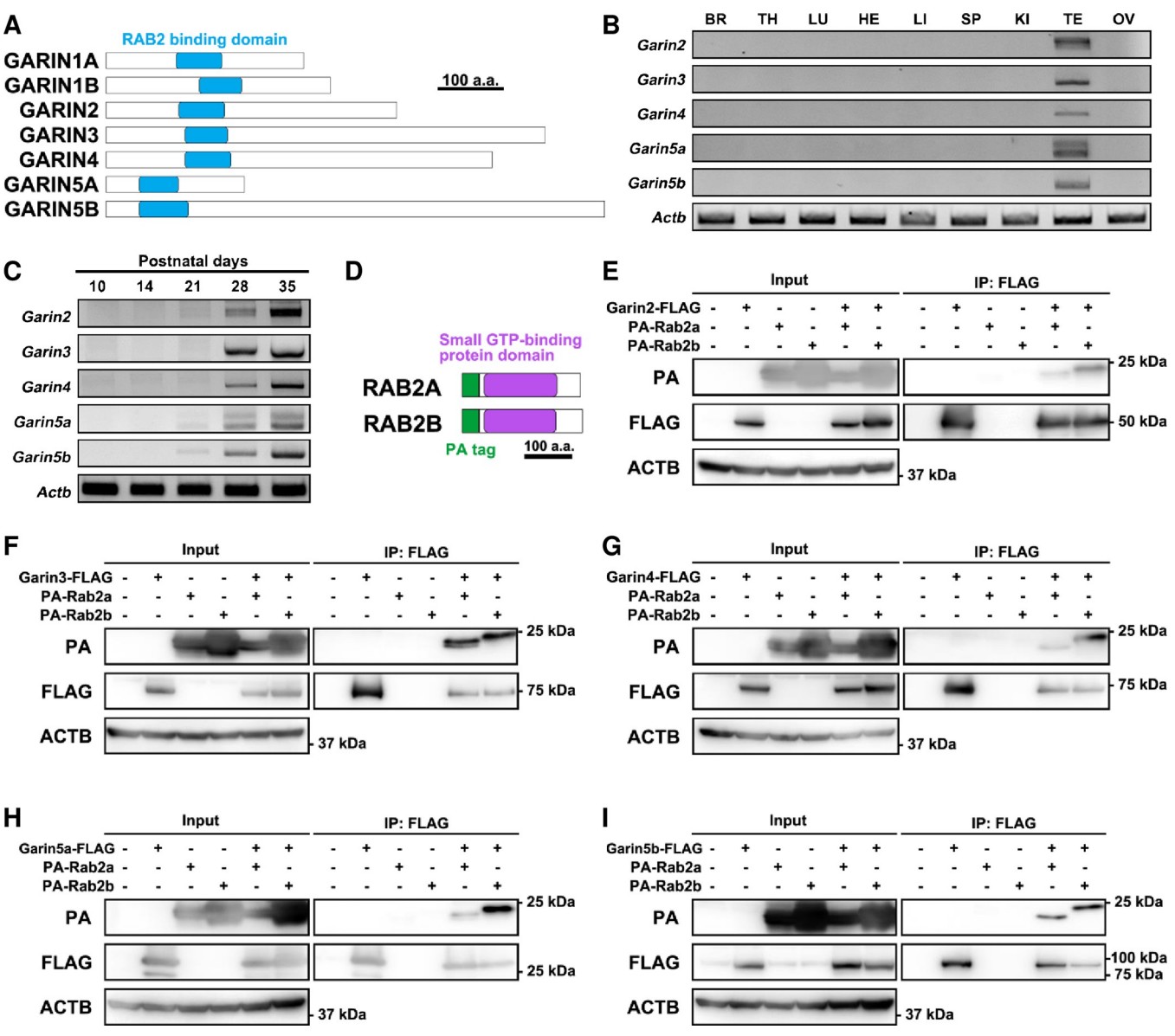

**Fig 1. GARINs show predominant expression in mouse testes and interact with RAB2A/B.** (A) Schematic representation of GARIN family members. The RAB2-binding domain is shown in blue. (B) RT-PCR for GARIN family members was performed with cDNA obtained from mouse tissues. *Actb* was used as a loading control. BR, brain; TH, thymus; LU, lung; HE, heart; LI, liver; SP, spleen; KI, kidney; TE, testis; OV, ovary. (C) RT-PCR was performed with cDNA obtained from postnatal testes with primers for GARIN family members. *Actb* was used as a loading control. (D) Schematic representation of the domains of RAB2A/B used in Fig 1E-I. The PA tag is shown in green and the Small GTP-binding protein domain is shown in magenta. (E—I) Co-IP of GARINs and RAB2A/B. Anti-FLAG and anti-PA antibodies were used to detect GARINs-FLAG and PA-RAB2A/B, respectively. In Fig 1E, the input PA bands were saturated because the signals were strong. Anti-ACTB antibody was used to detect ACTB as an endogenous control.

PCR product with the restriction enzyme BsrBI. No obvious abnormalities were observed in the development or behavior of all *Garin2-5b* KO mice.

We then caged wild-type (WT) female mice with individual male *Garin2-5b* KO or WT mice to test their *in vivo* fertility. WT females mated with WT males delivered 8.2 ± 2.7 pups on average from every coitus event. The fertility of *Garin2⁻/⁻* and *Garin4⁻/⁻* males seemed to be lower but were not statistically significant (6.5 ± 4.0 and 6.6 ± 3.8 pups/plug, respectively) compared to WT males. In contrast, *Garin5a⁻/⁻* and *Garin5b⁻/⁻* males were subfertile with 3.0 ± 4.1

and 3.8 ± 3.6 pups/plug, respectively. Furthermore, $Garin3^{-/-}$ males completely failed to sire pups although plugs were confirmed (Fig 2A). We also housed $Garin3^{-/-}$ males with superovulated WT females, collected eggs 8 hours after the mating, and confirmed that *Garin3* mutant spermatozoa cannot fertilize eggs *in vivo* (S3A and S3B Fig).

To find out the cause of impaired male fertility, we observed the testis and epididymis of *Garin2-5b* KO males. The testicular weights of *Garin2-5b* KO males were comparable with WT males (Fig 2B). Moreover, no overt abnormalities were found in testis or epididymis sections of *Garin2-5b* KO mice (Fig 2C). We then conducted *in vitro* fertilization (IVF) to further analyze the fertilizing ability of *Garin2-5b* KO spermatozoa. In the presence of cumulus cells and intact zona pellucida (ZP), the fertilization rates of each *Garin2-5b* KO spermatozoa were significantly decreased compared to WT mice (Fig 3A). Similarly, even though we removed cumulus cells before insemination, the fertilization rates of each *Garin2-5b* KO spermatozoa were lower than those of WT spermatozoa (Fig 3B). However, the decreased fertilization rates of each *Garin2-5b* KO spermatozoa were rescued by removing both the cumulus cells and ZP before insemination (Fig 3C). These results indicate that the ability to penetrate the ZP was impaired in all *Garin2-5b* KO spermatozoa *in vitro*.

### *Garin2*, *Garin3*, *Garin5a*, and *Garin5b* KO mice exhibit impaired sperm motility

We next examined sperm motility by computer-assisted sperm analysis (CASA) after 10 and 120 minutes incubation in a capacitation medium as defective motility could result in impaired ZP penetration [19]. Using the same CASA system, minor motility defects were found in spermatozoa with impaired ZP penetration [20]. For both 10 and 120 minutes, the percentages of motile spermatozoa were significantly decreased only in *Garin2* KO spermatozoa compared to WT spermatozoa (S4A and S4B Fig). Despite the percentages of motile spermatozoa being comparable in other *Garin* KO mice compared to WT mice, kinetic parameters such as straight line velocity (VSL), curvilinear velocity (VCL), and average path velocity (VAP) were reduced in *Garin3*, *Garin5a*, and *Garin5b* KO mice after 10 minutes of incubation (Fig 3D–3F). After 120 minutes of incubation, VSL was lower in *Garin5b* KO spermatozoa, VCL was lower in *Garin2*, *Garin3*, and *Garin5b* KO spermatozoa, and VAP was lower in *Garin2* and *Garin5b* KO spermatozoa (Fig 3G–3I). Although *Garin2*, *Garin3*, *Garin5a*, and *Garin5b* KO spermatozoa exhibited impaired motility, kinetic parameters of *Garin5b* KO spermatozoa (with the worst mean values) were still comparable with or higher than those of inbred C57BL/6J WT spermatozoa (S4C and S4D Fig), suggesting that the impaired ZP penetration cannot be explained only by diminished sperm motility.

### *Garin2-5b* KO male mice show abnormal sperm head morphology

To further analyze the cause of impaired fertility in *Garin2-5b* KO males, we observed spermatozoa obtained from the cauda epididymis and found that sperm heads of all *Garin2-5b* KO males exhibited abnormal morphology (Fig 4A). WT sperm heads showed a distinctively sharp and elongated hook; however, in *Garin2-5b* KO males, with visual examination, 88.4%, 91.5%, 92.4%, 47.0%, and 17.9% of the sperm heads exhibited bluntness in the hook regions, respectively (Fig 4A and 4B). Notably, in *Garin4* KO, approximately 33.1% of spermatozoa displayed one or multiple vacuoles in their heads (Figs 4A and S5).

To further compare the sperm head morphology of WT and *Garin2-5b* KO mice quantitatively, we conducted elliptic Fourier descriptors (EFDs) and principal component (PC) analysis on sperm heads to quantify the morphology [21], which have the advantage of being able to eliminate orientation errors caused by interference, image size, and starting point of the

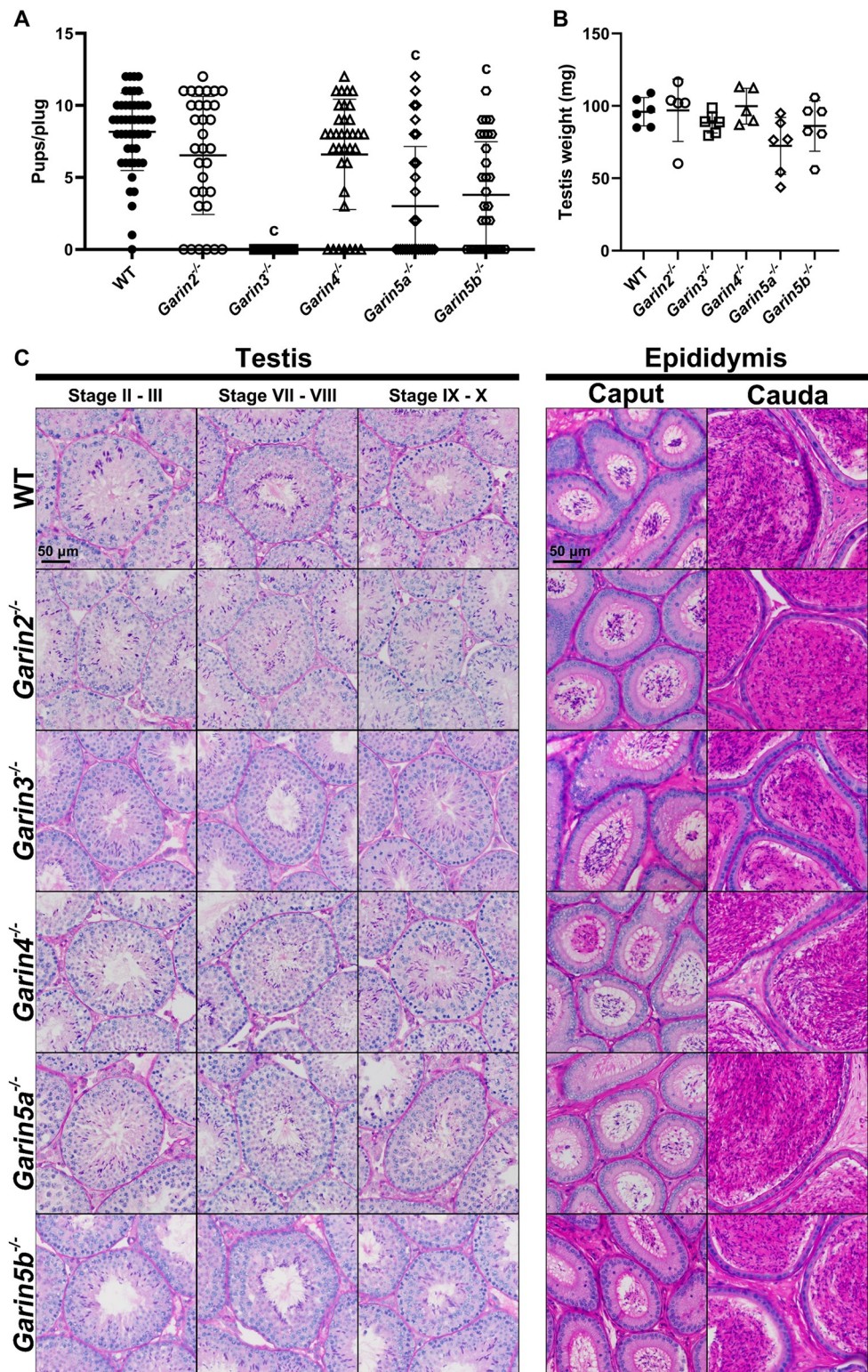

**Fig 2. *In vivo* fertility test and histological analyses of *Garin2-5b*$^{-/-}$ males.** (A) Pups per plug for WT males and *Garin2-5b* (*Garin2, 3, 4, 5a,* and *5b*) KO males. A vaginal plug was considered as a sign of coitus. For each genotype, 5 males were tested. Pups/plug of *Garin2-5b*$^{-/-}$ males were compared with WT males (One-way ANOVA, c, $P < 0.001$). (B) Testis weight of WT and *Garin2-5b*$^{-/-}$ males. The difference between testis weight of WT and *Garin2-5b*$^{-/-}$ males was not significant (One-way ANOVA, $P > 0.05$). (C) PAS-staining of testis and epididymis sections, from WT and *Garin2-5b*$^{-/-}$ males.

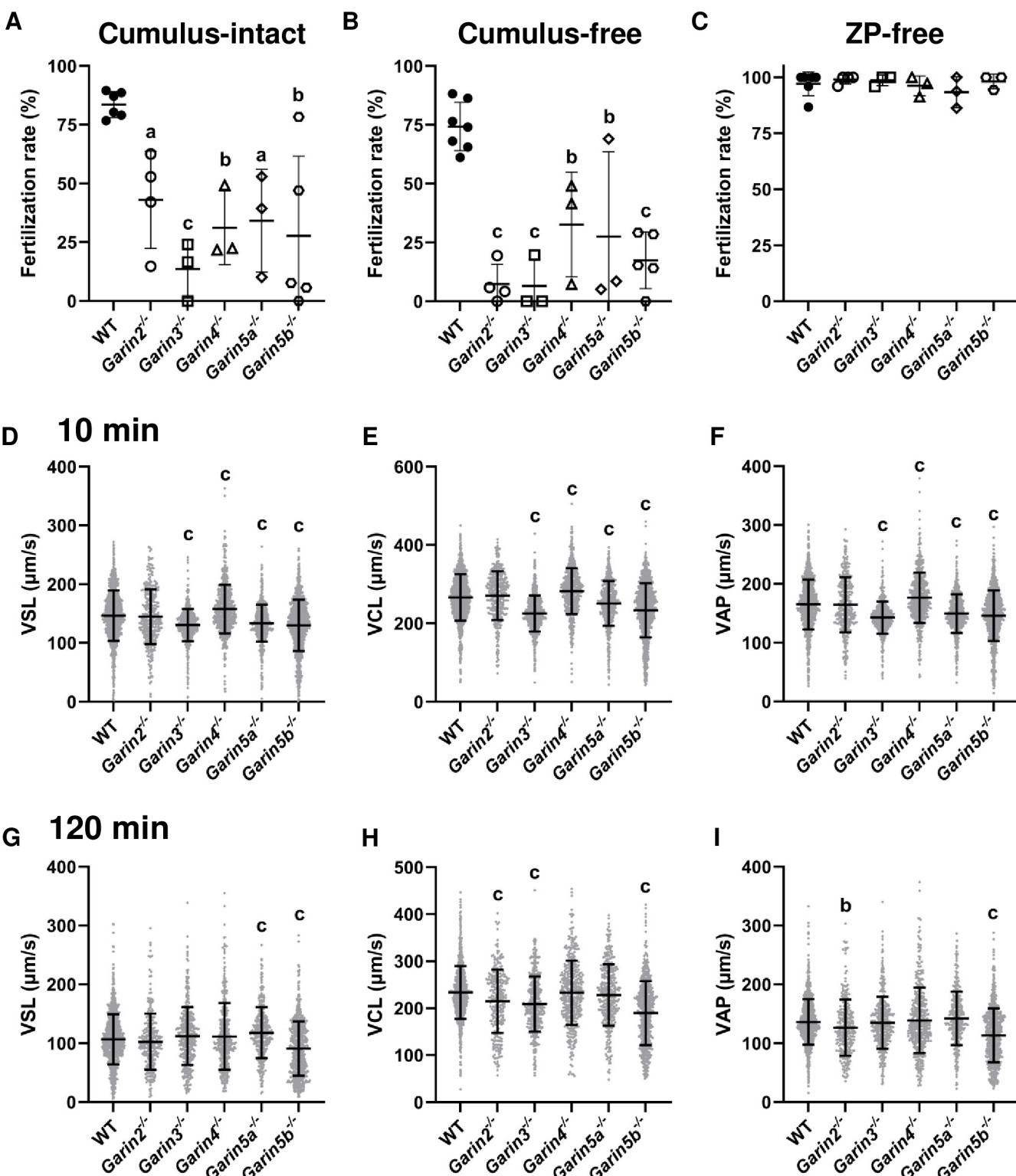

**Fig 3. *In vitro* fertilization and motility of *Garin2-5b* KO spermatozoa.** (A—C) *In vitro* fertilization analyses of WT and *Garin2-5b* (*Garin2*, *3*, *4*, *5a*, and *5b*) KO males in cumulus-intact condition (A), cumulus-free condition (B), and zona pellucida-free (ZP-free) condition (C). (D—I) straight line velocity (VSL), curvilinear velocity (VCL), and average path velocity (VAP) of WT spermatozoa and *Garin2-5b* KO spermatozoa after incubating for 10 minutes (D–F) or 120 minutes (G–I). Velocity parameters of *Garin2-5b* KO spermatozoa were compared with WT (One-way ANOVA, a, $P < 0.05$; b, $P < 0.01$; c, $P < 0.001$). For each KO mouse line and WT mouse, more than 3 males were analyzed.

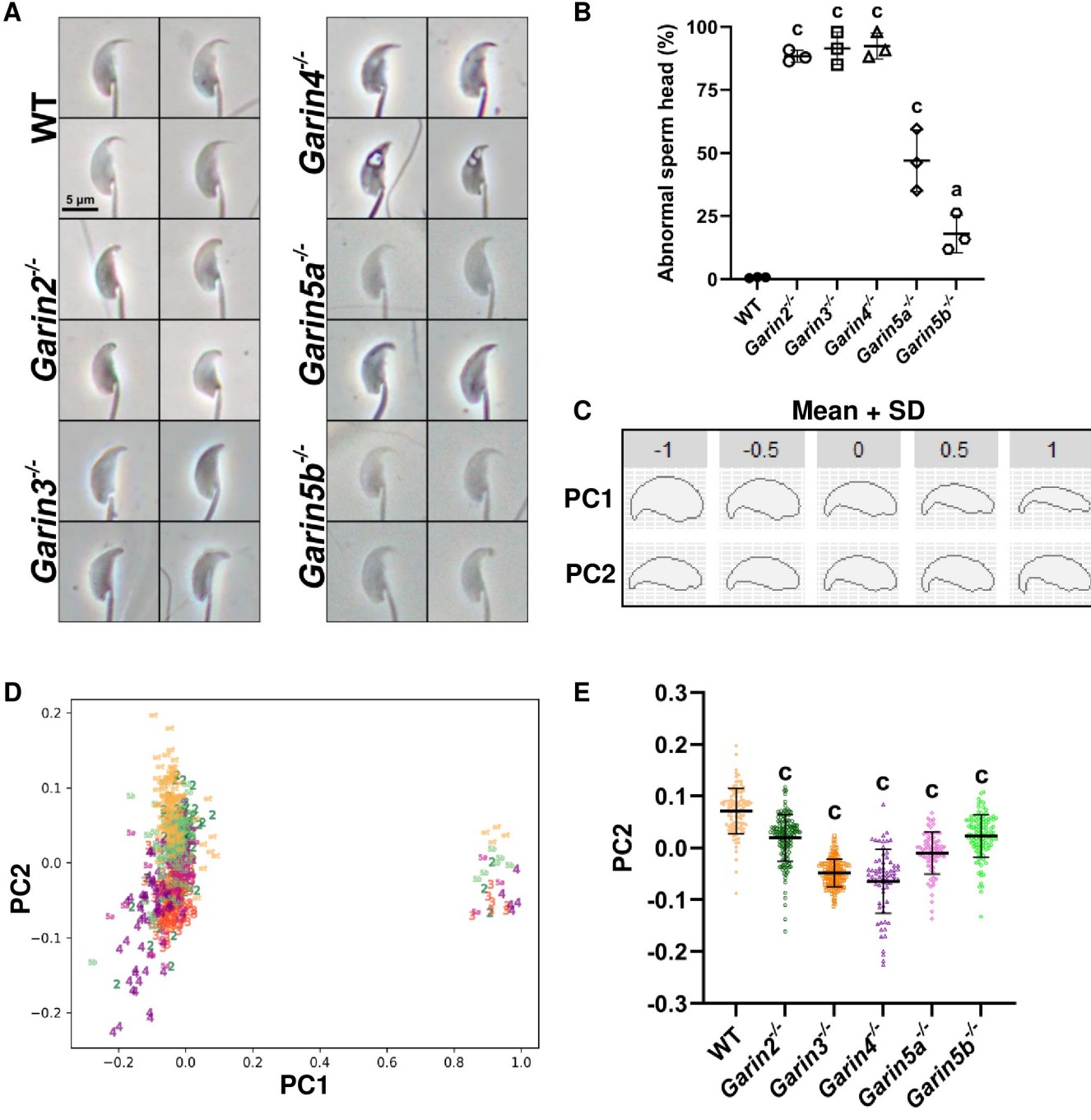

**Fig 4. Sperm head morphology examination by elliptic Fourier descriptors (EFDs) and principal component (PC) analysis.** (A) Representative images of sperm head morphology of WT and *Garin2-5b* (*Garin2*, *3*, *4*, *5a*, and *5b*) KO spermatozoa. (B) The percentage of abnormal sperm heads examined under a microscope. (C) Two PCs were generated against different aspects of the sperm heads. (D) PC1 and PC2 scores of WT and *Garin2-5b* KO sperm heads. PC2 distinguished sperm morphology but not PC1 between WT and *Garin2-5b* KO. (E) PC2 scores from (D) were plotted solely. *Garin2-5b⁻/⁻* groups were compared with the WT group (One-way ANOVA, a, $P < 0.05$; c, $P < 0.001$). For each KO mouse line and WT mouse, more than 3 males were analyzed.

contour [22]. Subsequently, we generated 2 PCs against different aspects of sperm head morphology (Fig 4C). Mean shape of each PC is generated as the average head shape of WT and *Garin2-5b* KO spermatozoa (Fig 4C). PC1 mainly features the width of the sperm head,

whereas PC2 focuses on the length and direction of the sperm hook. By separating the sperm head morphology by PC1 and PC2, we found that for the PC1 axis, data extracted by EFDs forms 2 clusters around -0.1 and +1.0, in contrast, data was evenly distributed from -0.2 to +0.2 on the PC2 axis, indicating that the morphology of *Garin2-5b* and WT spermatozoa was distinguished by PC2 but not PC1 (Fig 4D). Furthermore, all *Garin2-5b* KOs exhibited significantly lower values on PC2 (Fig 4E), indicating that each KO of *Garin2-5b* have shortened sperm hooks. These results suggest that the ZP penetration failure can be due to abnormal sperm head morphology in *Garin2-5b* KO mice. In particular, *Garin3* KO mice showed lower PC2 value with less variation (Fig 4E), which may be related to severe ZP penetration failure (Fig 3A) and infertility (Fig 2A).

## *Garin3* KO spermatozoa display abnormal acrosomal morphology

We have been shown that each KO of *Garin1a* and *Garin1b* results in abnormal sperm head and acrosomal morphology [13]. Therefore, we observed the acrosome of *Garin2-5b* KO spermatozoa with PNA staining. Although the majority of the acrosome from *Garin2-5b* KO spermatozoa seemed normal, some of the acrosomes of *Garin3* KO spermatozoa appeared to expand (Fig 5A), consistent with *Garin1a* and *Garin1b* KO spermatozoa [13]. We quantified the PNA-positive area in WT and *Garin2-5b* KO spermatozoa and found that the acrosome area of *Garin3* KO spermatozoa was significantly larger than that of WT spermatozoa (Fig 5B), suggesting that abnormal head morphology of *Garin3* KO spermatozoa may be caused by impaired acrosome biogenesis. In addition, we observed nuclear vacuoles in the Hoechst staining of the *Garin4* KO spermatozoa (Fig 5A), consistent with the phase contrast observation (Fig 4A).

Because acrosomal morphology was abnormal in *Garin3* KO spermatozoa, we measured acrosome reaction rates after 10 minutes and 4 hours incubation in a capacitation medium, and also after treatment with the $Ca^{2+}$ ionophore A23187 [23]. There were no significant differences in acrosome reaction rates between WT and *Garin3* KO spermatozoa in these conditions (Fig 5C), supporting the idea that the ZP penetration failure is caused by abnormal sperm head morphology but not impaired acrosome reaction.

## Deletion of *Garin3* affects sperm head plasma membrane proteins

To further analyze the effects of *Garin3* deletion, which caused the severest impairment in male fertility among *Garin2-5b* KOs, we performed mass spectrometry (MS) analyses of mature spermatozoa collected from cauda epididymis. We found that not only GARIN3 was completely absent in *Garin3* KO spermatozoa, but also 18 proteins such as ADAM4, ADAM6A, and ADAM6B were significantly downregulated in *Garin3* KO spermatozoa (S2 Table). We performed gene ontology (GO) analysis on downregulated proteins (Fig 6A) and found that proteins categorized in the sperm head plasma membrane and male gonad development were significantly downregulated in *Garin3* KO spermatozoa. It has been shown that the expression levels of ADAM4 and ADAM6 are dramatically reduced in *Adam2* KO and *Adam3* KO spermatozoa, and ADAM6 forms a complex with ADAM2 and ADAM3 [24]. Therefore, we analyzed the amounts of ADAM proteins for those antibodies that were available for us and found that ADAM2, ADAM3, and ADAM32 were downregulated in *Garin3* KO spermatozoa (Fig 6B). As a negative control, the amount of IZUMO1, a transmembrane protein localized in the acrosomal membrane, was not affected. These results suggest that not only head morphology but also the amounts of plasma membrane proteins were affected in *Garin3* KO spermatozoa. Moreover, in *Garin2-5b* KO spermatozoa, only *Garin3* KO

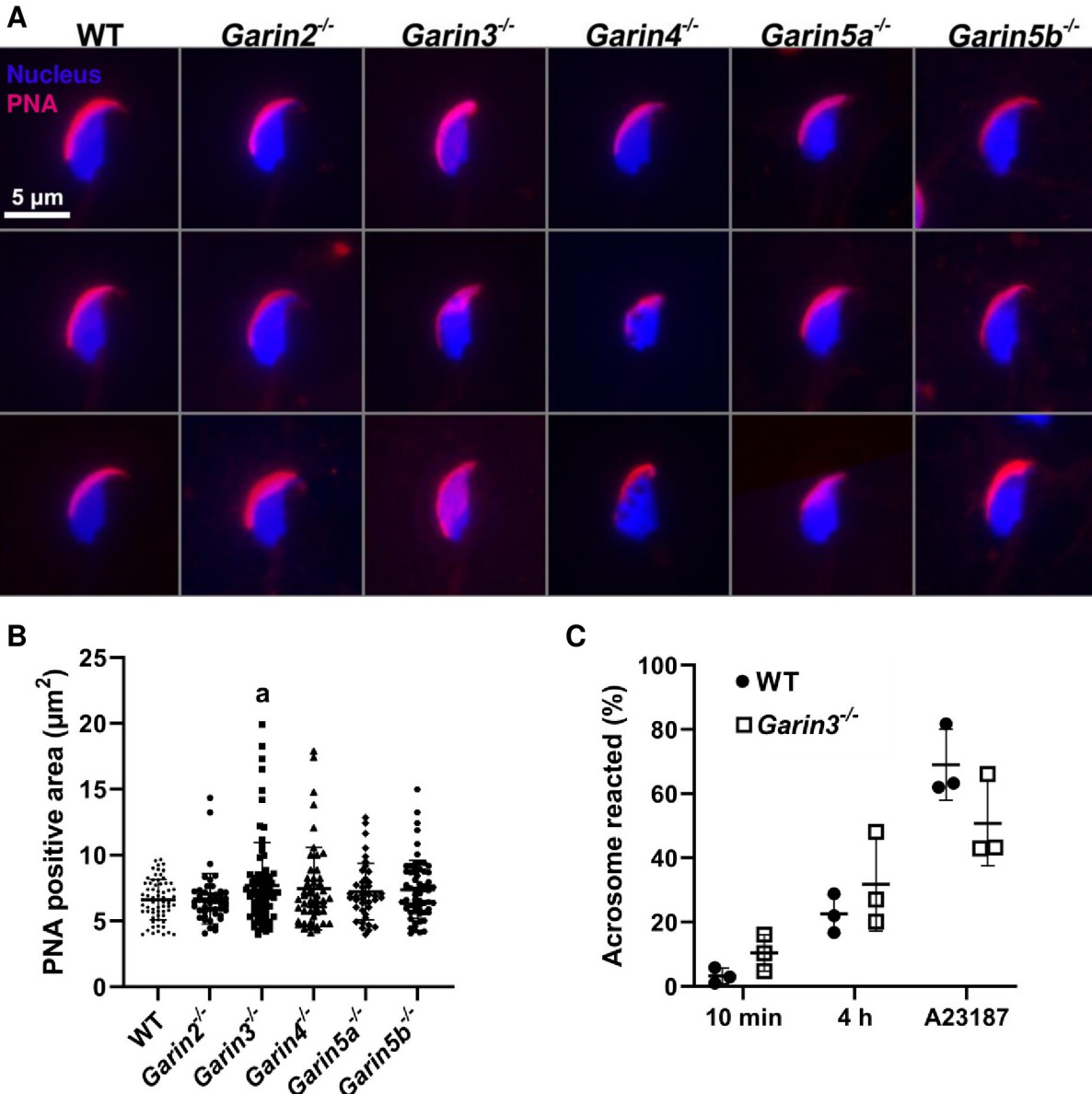

**Fig 5. Acrosome morphology of *Garin2-5b* KO spermatozoa and acrosome reaction of *Garin3* KO spermatozoa.** (A) Acrosomal morphology of WT and *Garin2-5b* (*Garin2*, *3*, *4*, *5a*, and *5b*) KO mice. The acrosome was stained with PNA (red) while nuclei were stained with Hoechst33342 (Blue). (B) PNA positive area of WT and *Garin2-5b* KO spermatozoa was quantified. *Garin2-5b* KO were compared with WT (One-way ANOVA, a, $P < 0.05$). For each KO mouse line and WT mouse, 3 males were analyzed. (C) Acrosome reaction rates of WT and *Garin3* KO spermatozoa after incubating for 10 minutes, 4 hours, and after adding $Ca^{2+}$ ionophore A23187. The difference between acrosome reaction rates of WT and *Garin3* KO mice in each condition was not significant (Student's t-test, $P > 0.05$).

spermatozoa showed downregulated ADAM3 (S6A Fig), suggesting the functional differences of GARIN3 among GARIN2-5B.

Because it has been shown that in the absence of ADAM2, ADAM3, or ADAM6, the ZP binding ability is severely impaired [25–27], we analyzed ZP binding ability of *Garin3* KO spermatozoa (Fig 6C). Although the ZP binding ability of *Garin3* KO spermatozoa is significantly reduced, 5.5 spermatozoa on average could still bind to the ZP, which is about 31.3% of the binding ability compared to WT spermatozoa (Fig 6C and 6D). These results suggest that

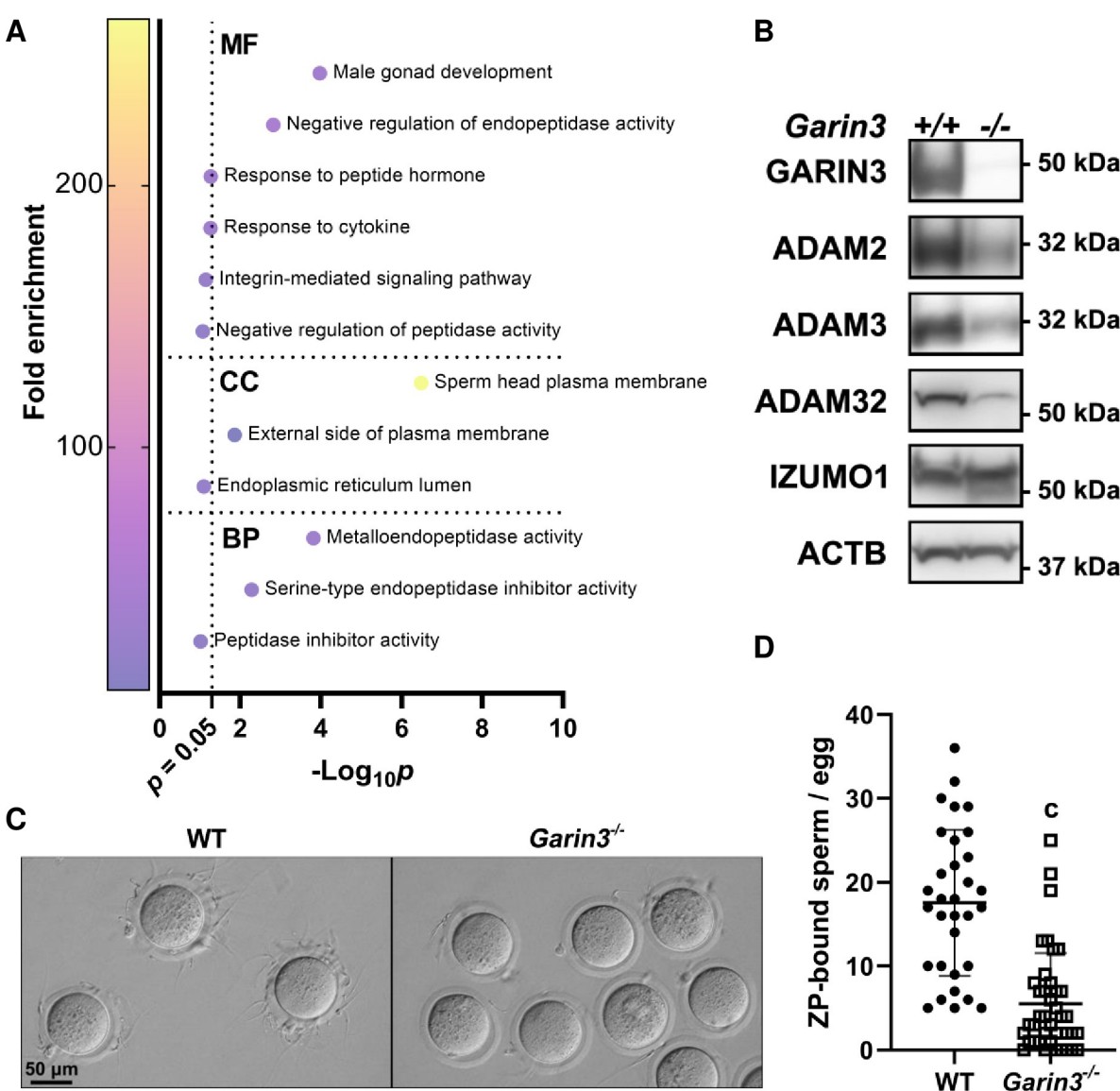

**Fig 6. KO of *Garin3* led to the loss of ADAMs and aberrant ZP-binding ability.** (A) GO analysis of proteins significantly downregulated in *Garin3* KO spermatozoa (S2 and S6 Tables). MF, CC, and BP are abbreviations for molecular function, cellular component, and biological process, respectively. (B) Western blotting analyses revealed the amounts of ADAM2, ADAM3, and ADAM32 decreased in *Garin3* KO spermatozoa. IZUMO1 and ACTB were detected as endogenous controls. (C) Representative image of ZP-binding analyses of WT and *Garin3* KO spermatozoa. (D) Numbers of ZP-bound spermatozoa per egg. *Garin3* KO spermatozoa exhibited lower ZP-binding ability compared to WT spermatozoa. For *Garin3*[-/-] and WT mice, 3 males were used for the ZP-binding assay (Student's t-test, c, $P < 0.001$).

the combinatorial effect of impaired sperm head morphology and ZP binding ability resulted in ZP penetration failure in *Garin3* KO spermatozoa.

## GARIN3 binds to both inactive and active forms of RAB2A/B

To investigate the cause of abnormal acrosome morphology in *Garin3* KO spermatozoa, we analyzed proteins involved in acrosome formation. We found that the amounts of GOPC, ZPBP1, and SPACA1 were not affected in *Garin2-5b* KO testes (S6B Fig). Our previous study on another GARIN protein family member, GARIN1B, demonstrated that N-linked

glycosylation of SPACA1 in the *Garin1b* KO testis was impaired [13]. In contrast, WB using an anti-SPACA1 antibody showed normal bands for *Garin3* KO testis (S6B Fig). This result suggests that although GARIN1B and GARIN3 are members of the same protein family and deletion of either gene caused sperm head malformation and infertility, their functions could be different.

RAB2A/B is a GTPase that is only active in the GTP-bound form and is inactive in the GDP-bound form [28]. RAB2A/B can be mimicked into constitutively active (CA; GTP-bound) and constitutively negative (CN; GDP-bound) forms, by introducing the Q65L and S20N amino acid substitutions, respectively [29]. One feature of GARIN1B is that it only interacts with the CA form but not with the CN form of RAB2A/B. In contrast, co-IP analyses using HEK293T cells showed that GARIN3 could bind to both CA and CN forms of RAB2A/B (Fig 7A and 7B). Moreover, further analyses showed that, among GARIN2, GARIN4, GARIN5A, and GARIN5B, GARIN5A could bind to both CA and CN forms of RAB2A although GARIN2/4/5B could bind to the CA form strongly and bind to the CN form weakly (S6C Fig). Moreover, GARIN5B could bind to both the CA and CN forms of RAB2B, but only GARIN2/4/5A binds to the CA form of RAB2B preferentially (S6D Fig). These results suggest that GARIN3, GARIN5A, and GARIN5B have different characteristics among all members of the GARIN protein family.

Previously, we have shown that GARIN1B is localized in the Golgi-apparatus when expressed in COS-7 cells [13]. In contrast, we found that GARIN3 was localized in the cytoplasm of COS-7 cells (Fig 7C). However, when RAB2A or RAB2B was co-expressed with GARIN3, GARIN3 was localized in the Golgi-apparatus, suggesting that RAB2A/B recruits GARIN3 to the Golgi-apparatus. It has also been shown that RAB2B recruits GARIN5A to the Golgi apparatus [30]. These results also indicate that GARIN3 and GARIN5A have different characteristics from that of GARIN1B.

## Discussion

Consistent with GARIN1A and GARIN1B, we found other members of the GARIN family, GARIN2, GARIN3, GARIN4, GARIN5A, and GARIN5B, could interact with RAB2A/B that was suggested to function in acrosome biogenesis in several species [13,31,32]. Further, RT-PCR analyses revealed that *Garin2-5b* were expressed predominantly in testes. We generated *Garin2⁻/⁻*, *Garin3⁻/⁻*, *Garin4⁻/⁻*, *Garin5a⁻/⁻*, and *Garin5b⁻/⁻* mice, and found that *Garin3⁻/⁻* male mice were infertile and *Garin5a⁻/⁻* and *Garin5b⁻/⁻* male mice were subfertile. Although *Garin2⁻/⁻* and *Garin4⁻/⁻* male mice were fertile, ZP penetration ability was shown to be reduced in all *Garin2-5b⁻/⁻* mice using IVF. All *Garin2-5b* KO spermatozoa showed impaired head morphogenesis, which may result in impaired ZP penetration in combination with reduced sperm motility.

Sperm head formation is important for male fertility [33–35]. Because each mutant of *Garin2-5b* causes sperm head malformation, we performed EFDs and PC analysis to quantify head morphology. EFD was first introduced in 1982, and it is used to describe the morphology by generating a series of ellipses that approximate the shape [36], which has been applied to evaluate various biological samples [21,37]. The analyses showed that PC2 values that focused on the length and direction of the sperm hook were lower in all *Garin2-5b* KO mice compared to WT mice. Indeed, *Garin4* KO spermatozoa show the lowest average PC2 value among all of the *Garin2-5b* KO spermatozoa; however, the standard deviation was also the largest. Spermatozoa with high PC2 values in *Garin4* KO mice may be responsible for better fertilizing ability in both *in vitro* and *in vivo* compared to *Garin3* KO mice. EFDs and PC analysis can be a powerful tool for quantifying sperm head morphology, which may also be used to diagnose teratozoospermia, a condition of infertile men who exhibit abnormally shaped spermatozoa.

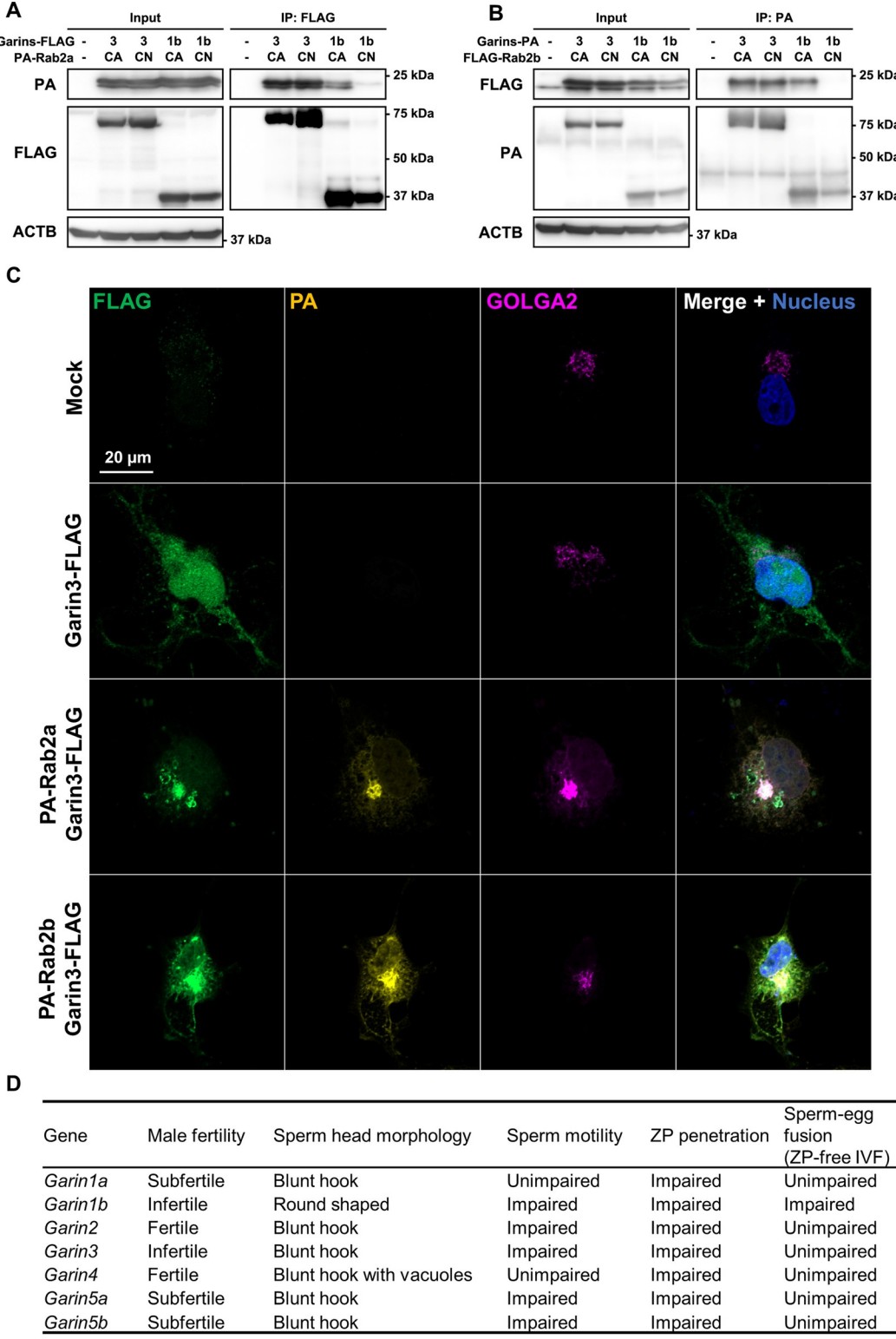

**Fig 7. Differences in the interaction between GARIN1B/3 and RAB2A/B.** (A and B) Co-IP of GARIN1B/3 and constitutively active (CA)/constitutively negative (CN) forms of RAB2A/B. GARIN3 interacts with both CA and CN forms of RAB2A/B, while GARIN1B only interacts with the CA form of RAB2A/B. (C) Localization of GARIN3-FLAG (green) with or without PA-RAB2A/B (yellow) in COS-7 cells. The Golgi apparatus and nuclei were stained with anti-GOLGA2 (magenta) and Hoechst33342 (Blue), respectively. (D) Summary of phenotypes observed on male reproduction for all *Garins* KO male mice.

*Garin1b* KO mice exhibit sperm head malformation, abnormal *N*-linked glycosylation of SPACA1, defective acrosome biogenesis with impaired acrosome reaction, and male sterility [13]. In contrast, the mutation in *Garin3* causes sperm head malformation and male sterility, but does not cause severely impaired acrosome biogenesis or impaired acrosome reaction. Moreover, *Garin3* KO spermatozoa exhibited normally processed SPACA1. Instead, the amounts of ADAM2, ADAM3, and ADAM6, which are crucial for sperm-ZP binding and male fertility [25–27], and the amounts of other ADAM family members, ADAM4 and ADAM32, were decreased in *Garin3* KO spermatozoa. Among these ADAM family members, it has been suggested ADAM2-ADAM3-ADAM4, and ADAM2-ADAM3-ADAM6 may form complexes [38]. GARIN3 may interact with some proteins of the complexes as well as ADAM32 to regulate the processing of these membrane proteins. Notably, despite the decreased amount of ADAMs, *Garin3* KO spermatozoa still maintain 31.3% ZP binding ability compared to WT spermatozoa. Because a transgenic study has shown that a small amount of ADAM3 is able to fully rescue the fertility of *Adam3* KO male mice [39], reduced amounts of ADAMs alone may not be sufficient to exterminate male fertility, and the shape of the sperm head may also be involved.

RABs are involved in membrane trafficking and vesicle fusion/fission by interacting with numerous proteins including its effector which interacts with the active form of RABs [40,41]. Indeed, GARIN1B and GARIN3 possess a RAB2-binding domain in the N-terminus and bind to RAB2A/B, however, in the WB study, we show that GARIN3 interacts with both the CA and CN forms of RAB2A/B in HEK293T cells (Fig 7A and 7B), and GARIN3 is present in mature spermatozoa (Fig 6B). Given the fact that GARIN1B only interacts with the CA form of RAB2A/B and was absent in mature spermatozoa [13], GARIN3 may function differently from GARIN1B. This difference is possibly due to the varied length and low similarity of the regions other than the RAB2-binding domain. It should be noted that we analyzed the interaction and localization of GARIN2-5B and RAB2A/B using a heterologous expression system as we have no functional GARIN2-5B antibodies for IP or immunofluorescence. Further analyses of their interaction and co-localization in testes would reveal the functional relationship of GARIN2-5B and RAB2A/B. Furthermore, studying the binding affinity of GARINs and the CA and CN forms of RAB2A/B using surface plasmon resonance or biolayer interferometry would reveal the binding specificity of each GARINs with RAB2A/B in more detail.

Our previous study indicates that RAB2A/B is localized in the Golgi apparatus of the round spermatids and localized in the acrosome of the elongating spermatids and that GARIN1B may assist RAB2A/B in fulfilling their vesicle transportation from the Golgi apparatus to the acrosome [13]. GARIN3 may have similar functions to some extent since *Garin3* KO mice show morphological abnormalities in the acrosome and sperm heads, although not as severe as in *Garin1b* KO mice. However, in the present study, we show that RAB2A/B recruited GARIN3 to the Golgi-apparatus (Fig 7C), and the amounts of membrane proteins, namely ADAMs, were decreased in *Garin3* KO spermatozoa. Taken together, GARIN3 may also function in a RAB2A/B-dependent manner in the Golgi apparatus for membrane protein trafficking, vesicle anchoring, and/or processing. KO of *Garins* affects male fertility and sperm head morphogenesis to various extents (Fig 7D), notably, *Garin4* KO sperm displays vacuoles in their head. Therefore, other than GARIN3, some members of the GARIN protein family may also possess unique functions that are different from other GARINs.

Among GARIN2-5B, we demonstrated that GARIN3 is essential for male fertility, and all GARIN2-5B are indispensable for sperm head morphogenesis. GARINs are all conserved in humans and GARIN2 is found under positive selection in Europeans [42]. Therefore, GARINs may play important roles in humans, and inactivation of them may lead to teratozoospermia. Moreover, it is known that nuclear vacuoles are present in the spermatozoa of fertile and

infertile men and related to male fertility potential, however, the origin and the formation process of nuclear vacuoles are not well characterized [43]. Further analysis of GARINs may lead us to a better understanding of sperm head morphogenesis, teratozoospermia, and male infertility.

## Materials and methods

### Ethics statement

Mouse experiments were approved by the Animal Care and Use Committee at the Research Institute for Microbial Diseases, Osaka University (#Biken-AP-H30-01).

### Animals

Mice were purchased from CLEA Japan (Tokyo, Japan) or Japan SLC (Shizuoka, Japan). All mice were maintained under specific-pathogen-free conditions with *ad libitum* feeding, under an artificial 12-h light/12-h dark cycle. All gene-modified mice generated in this study will be available through either the RIKEN BioResource Research Center or the Center for Animal Resources and Development (CARD), Kumamoto University.

### Generation of plasmids

The open reading frames (ORFs) of *Garin2*, *Garin3*, *Garin4*, *Garin5a*, and *Garin5b* were cloned, amplified, and inserted into the pCAG1.1 vector (Addgene, Plasmid, #173685) with a FLAG-tag or PA-tag. Plasmids encoding GARIN1B and CA/CN forms of RAB2A/B were described previously [13]. Further, Dr. Mitsunori Fukuda (Graduate School of Life Sciences, Tohoku University, Tohoku, Japan) kindly provided plasmids encoding the CA/CN form of RAB2B [29].

### Transfection of HEK293T cells and COS-7 cells

HEK293T and COS-7 cells were maintained with Dulbecco's Modified Eagle Medium (DMEM; Thermofisher, Waltham, MA) with 10% fetal bovine serum (FBS; Sigma-Aldrich, St. Louis, MO) and 1% penicillin/streptomycin (Thermofisher) at 37°C under 5% $CO_2$. Plasmids were transfected by the calcium phosphate transfection method and by polyethyleneimine (PEI) for HEK293T cells and COS-7 cells, respectively. Cells were harvested 16 hours after transfection.

### Protein extraction of HEK293T cells, mouse testes, and spermatozoa

HEK293T cells were harvested with 1% Triton lysis buffer (1% Triton X-100, 50 mM Tris-HCl pH 7.5, 150 mM NaCl) with 1% protease inhibitor cocktail (Nacalai Tesque, Kyoto, Japan) and incubated for 2 hours on ice. Mouse testes and epididymis were dissected by forceps and ophthalmologic scissors under a dissection microscope. Then, seminiferous tubules inside the testes were pulled out and homogenized in 1 mL urea lysis buffer (6 M urea, 2 M thiourea, 2% sodium deoxycholate) with a homogenizer. Mature spermatozoa were extracted from the cauda epididymis and resuspended in 1 mL phosphate-buffered saline (PBS). After centrifuge at $300 \times g$ at 4°C for 10 minutes, the supernatant was discarded and then resuspended with 0.1 mL urea lysis buffer. Both testes and sperm lysates were incubated on ice for 2 hours. The lysate was then centrifuged at $15,000 \times g$ at 4°C for 20 minutes and the supernatant was subjected to immunoprecipitation or Western blotting.

### Co-immunoprecipitation (Co-IP)

The supernatant obtained from HEK293T cells was subjected to Co-IP. Co-IP was performed using the Invitrogen Dynabeads Magnetic Beads (Thermofisher) under the protocol from the manufacturer. Antibodies used in this study are indicated in S3 Table.

### Antibody generation

Rabbit polyclonal anti-GARIN3 antibody was produced by immunization with mouse GARIN3 polypeptide (C plus SGKTREDKGKGHGRLRGKR). GARIN3 antibody was then purified using the GARIN3 polypeptide and SulfoLink coupling resin (Thermofisher).

### Western blotting (WB)

Protein extracted from mouse tissues and HEK293T cells were incubated at 95˚C in the SDS-sample buffer containing β-mercaptoethanol for 5 minutes. Next, samples were loaded on SDS-PAGE gels (ATTO Corp., Osaka, Japan), and then transferred to polyvinylidene fluoride (PVDF) membranes by using the Trans-Blot Turbo system (BioRad, Munich, Germany). Membranes were then blocked with 5% skim milk (BD Bioscience, Sparks, MD) in TBS-T (0.05 M Tris-HCl, 0.15 M NaCl, 0.05% Tween 20) for 1 hour at room temperature. Subsequently, membranes were incubated at 4˚C overnight with primary antibodies diluted in 5% skim milk in TBS-T. After 3 quick washes with TBS-T, membranes were further incubated with secondary antibodies diluted in 5% skim milk in TBS-T for 1 hour at room temperature. After 3 quick washes by TBS-T, membranes were treated with Amersham ECL Western Blotting Reagent (Cytiva, Tokyo, Japan) for 1 minute, and bands were detected with Amersham ImageQuant 800 (Cytiva). The dilution of primary antibodies used in this study is indicated in S3 Table. Anti-ZPBP1 antibody was a gift from Dr. Martin M. Matzuk [35].

### Sequence comparison and phylogenetic analysis

The sequences were obtained from the Ensembl database [44], and Clustal Omega was used for multiple sequences comparison and alignment [15]. Alignment was visualized using Jalveiw [45]. The identifier of sequences was specified in S4 Table.

### Reverse transcription polymerase chain reaction (RT-PCR)

Under the manufacturer's protocol, RNA was extracted from various tissues and postnatal testes of C57BL/6J mice by Trizol (Thermofisher) and then converted to cDNA with the SuperScript III First Strand Synthesis Kit (Thermofisher). Then, cDNA was used to analyze the expression of genes by PCR with KOD Fx Neo DNA Polymerase (Toyobo, Tokyo, Japan). The sequences of primers used for RT-PCR are listed in S5 Table.

### *In silico* gene expression analysis

The tissue-specific expression of genes was evaluated based on the data obtained from the Expression Atlas [18]. Data on the mRNA expression at different stages of spermatogenesis were used to validate the gene expression in male germ cells [17].

### Generation of *Garin2-5b* Knockout (KO) mice

A pair of guide RNAs (gRNAs) were designed to remove large parts of each coding sequence for *Garin2*, *Garin3*, *Garin4*, and *Garin5b*. For the coding sequence of *Garin5a*, one gRNA was designed to cause a nonsense mutation. The sequences of gRNAs are listed in S5 Table.

B6D2F1 females at 7-weeks-old were superovulated with 0.1 mL CARD HyperOva (Kyudo, Saga, Japan) and followed by injection of 5 units human chorionic gonadotropin (hCG; ASKA Pharmaceutical, Tokyo, Japan) after 48 hours. Females were then mated with B6D2F1 males and sacrificed to collect the 2 pronuclei (2PN) oocytes. Electroporation was performed to introduce the complex of CAS9, CRISPR RNA (crRNA), and trans-activating crRNA (tracrRNA) into 2PN oocytes [46]. Next, the oocytes were cultured in potassium simplex optimization medium (KSOM) to the 2-cell stage [47] and then transplanted into the oviducts of 0.5-day pseudopregnant ICR females. The F0 pups were naturally delivered or obtained by Cesarean section. For the F0 and F1 generations, the deletion of coding sequences was confirmed by genomic PCR and Sanger sequencing. Genotyping of F2 and later generations was only performed by genomic PCR. For $Garin5a^{-/-}$ mice, the genotyping was performed by BsrBI digestion of the PCR product. KOD Fx Neo DNA polymerase was used for PCR. The primers used in genomic PCR are shown in S5 Table.

### *In vivo* fertility test

Each KO or WT male was caged with three 7-week-old females for 8 weeks. During these 8 weeks, vaginal plugs were checked as a sign of successful coitus, and the number of pups born was counted. After 8 weeks, males were removed from the cage, then the number of pups was counted for 3 weeks as the females could give birth to the last litters.

### *In vitro* fertilization (IVF)

Female mice were superovulated with CARD HyperOva and hCG as previously stated before IVF. Male mice were dissected and mature spermatozoa were collected from cauda epididymis, then spermatozoa were released in Toyoda Yokoyama Hoshi (TYH) medium and incubated for 2 hours in the condition of 37˚C, 5% $CO_2$, to induce capacitation [48]. During these 2 hours, oocytes were extracted from the oviductal ampulla, and a portion of them was treated with 330 μg/mL of hyaluronidase (Wako, Osaka, Japan) or 1 mg/mL of collagenase (Sigma-Aldrich) to remove cumulus cells or the ZP, respectively. Then, spermatozoa were incubated with the oocytes at a concentration of $2 \times 10^5$ sperm/mL in 0.1 mL TYH drops. After 6 hours of insemination, 2PN embryos were counted and the rate of 2PN formation was determined as the fertilization rate.

### Acrosome reaction analysis

Cauda epididymal spermatozoa were released in TYH drops and incubated at 37˚C, 5% $CO_2$, for 10 minutes or 4 hours. After incubation, a portion of spermatozoa was spread onto glass slides and air-dried. Another portion was treated with A23187 (Merck Millipore, Darmstadt, Germany) in the TYH drops to induce the acrosome reaction for 10 minutes, then spermatozoa were spread onto glass slides and air-dried. 4% paraformaldehyde (PFA) in PBS was added to the glass slides for 10 minutes. After 3 quick washes with PBS, the slides were blocked with a blocking buffer consisting of 5% bovine serum albumin (BSA) and 10% goat serum in PBS for 1 hour at room temperature, then treated with anti-IZUMO1 antibody [49] in blocking buffer overnight at 4˚C. After 3 washes with PBS, the samples were incubated with fluorophore-conjugated secondary antibody for 1 hour at room temperature, washed 3 times with PBS, then incubated with 0.02% Hoechst 33342 (Thermofisher) in PBS for 10 minutes. Following 3 washes, slides were mounted using Immu-Mount (Thermofisher) before being observed with an Olympus BX-53 microscope (Tokyo, Japan).

## Sperm motility analysis

Spermatozoa were released from cauda epididymis into a 0.1 mL TYH drop, then after 10 minutes and 2 hours of incubation at 37˚C, 5% $CO_2$, sperm motility was measured with the CEROS II sperm analysis system (Hamilton Thorne Biosciences, Beverly, MA) [50].

## Elliptic Fourier descriptors (EFDs) and principal component (PC) analysis

Mature spermatozoa were extracted from the cauda epididymis and resuspended in 1 mL PBS. Then, spermatozoa were spread onto glass slides and images were captured under an Olympus BX-53 microscope. To represent sperm head shape with elliptic Fourier series, we extracted two-dimensional morphological information of sperm head contour to the coordinates, then used elliptical parameters to reproduce the curve of contour based on X and Y coordinates. In this study, we employed 20 elliptic harmonics to represent sperm head shape with 80 coefficients. Next, principal component analysis capturing the variation in multidimensional data and compressing it into a smaller number of principal components was applied to 80 coefficients. A detailed description focused on the mathematical perspective of EFDs and PC analysis on sperm heads was described by Mashiko et al. [21].

## Staining of acrosome by PNA

Mature spermatozoa were extracted from the cauda epididymis and resuspended in 1 mL PBS. Then, spermatozoa were spread onto glass slides and air-dried. After 10 minutes fixation by 4% PFA in PBS, slides were blocked with 5% BSA in PBS blocking buffer. After blocking, slides were probed with Alexa Fluor 568 conjugated Lectin PNA (Thermofisher). Following 3 washes and incubation with 0.02% Hoechst 33342 in PBS for 10 minutes, slides were washed 3 times with PBS and then mounted using Immu-Mount. Slides were observed with an Olympus BX-53 microscope.

## Histology analysis of testis and epididymis

Histological analysis was performed as described previously [51]. Bouin's fluid (Polysciences, Inc., Warrington, PA) was used to fix testes and epididymis, then, tissues were embedded in cassettes with paraffin and sectioned to 5 μm. PAS staining was performed on the sections, with periodic acid (Nacalai Tesque) and Schiff's reagent (Wako). Finally, slides were counterstained by hematoxylin, and images were captured with an Olympus BX-53 microscope.

## Mass spectrometry (MS) and gene ontology (GO) analysis

Proteins extracted from epididymal spermatozoa using the urea lysis buffer were subjected to nanocapillary reversed-phase liquid chromatography (LC)-MS/MS analysis with a C18 column (IonOpticks, Victoria, Australia) equipped on a nanoLC system (Bruker Daltonics, Billerica, MA) connected to a timsTOF Pro mass spectrometer (Bruker Daltonics) and the CaptiveSpray nano-electrospray ion source (Bruker Daltonics). The raw data was processed using DataAnalysis (Bruker Daltonics), and proteins were identified using MASCOT (Matrix Science, Tokyo, Japan) by recruiting the SwissProt database. Finally, quantitative value and fold change were calculated using Scaffold5 (Proteome Software, Portland, OR). The result of the MS study is indicated in S2 Table. Proteins significantly decreased in *Garin3* KO spermatozoa were subjected to GO analysis with DAVID Bioinformatics Resources [52,53]. Raw data of GO analysis was shown in S6 Table. Unadjusted p-values were plotted in Fig 6A.

## Sperm-ZP binding assay

As described above in the methodology of IVF, cumulus-free oocytes were inseminated with capacitated spermatozoa at a concentration of $2 \times 10^5$ sperm/mL in 0.1 mL TYH drops. After 30 minutes, oocytes were removed from TYH drops and fixed with 0.1% PFA in TYH for 10 minutes. Oocytes were then moved to a new TYH drop, and then images were captured using an Olympus IX73 microscope.

## Immunocytochemistry of COS-7 cells

COS-7 cells were fixed with 4% PFA in PBS for 15 minutes, then blocked with 3% BSA in PBS blocking buffer. After blocking, slides were incubated with primary antibodies diluted in blocking buffer overnight at 4˚C. After 3 quick washes with PBS, slides were probed with fluorescence protein conjugated secondary antibodies diluted in blocking buffer for 1 hour at room temperature. After secondary antibody incubation, the slides were incubated with 0.02% Hoechst 33342 in PBS for 10 minutes, washed 3 times with PBS, and then mounted using Immu-Mount. Slides were observed with a Nikon Eclipse Ti microscope.

## Statistical analysis

Data are presented as mean ± standard deviation (SD). Data consisting of two groups was statistically analyzed with the unpaired two-tailed Student's t-test. For data consisting of more than two groups, data was analyzed by one-way ANOVA. Statistical significance was defined as $P < 0.05$ (a, $P < 0.05$; b, $P < 0.01$; c, $P < 0.001$).

## Supporting information

**S1 Fig. Sequence comparison and expression of GARINs.** (A) Sequence comparison of GARINs in mice. (B–C) Phylogenetic analyses of amino acid sequences on mouse GARINs (B) or human GARINs (C). (D–E) Phylogenetic analyses of coding sequences on mouse *Garins* (D) or human *GARINs* (E). (F) Expression of GARINs and RAB2A/B in mouse tissues. Data was obtained from the Expression Atlas. NE, no expression. (G) Data on the mRNA expression of GARINs and RAB2A/B at different stages of spermatogenesis.
(TIF)

**S2 Fig. Generation of *Garin2-5b*<sup>-/-</sup> mice.** (A–E) KO mouse generation strategy, deletion size, deletion site, and genomic PCR of *Garin2* (A), *Garin3* (B), *Garin4* (C), *Garin5a* (D), and *Garin5b* (E).
(TIF)

**S3 Fig. Fertilization rates of *Garin3*<sup>-/-</sup> males.** (A) Representative image of eggs collected from WT females that were mated with WT or *Garin3*<sup>-/-</sup> males. Eggs were collected 8 hours after mating. (B) 2PN formation rates of eggs collected from WT females which were mated with WT or *Garin3*<sup>-/-</sup> males (Student's t-test, c, $P < 0.001$).
(TIF)

**S4 Fig. Sperm motility of *Garin2-5b*<sup>-/-</sup> mice.** (A–B) Motile sperm rates of WT and *Garin2-5b* (*Garin2*, *3*, *4*, *5a*, and *5b*) KO mice after incubating for 10 minutes (A) and 120 minutes (B). *Garin2-5b*<sup>-/-</sup> groups were compared with the WT group (One-way ANOVA, c, $P < 0.001$). (**C —D**) Average of straight line velocity (VSL), curvilinear velocity (VCL), and average path velocity (VAP) of WT spermatozoa [from C57BL/6 (B6) mice] and *Garin5b* KO spermatozoa [from C57BL/6 × DBA/2 (B6D2) hybrid mice] after incubating for 10 minutes (C) and 120

minutes (D) (Student's t-test, a, $P < 0.05$).
(TIF)

**S5 Fig. Vacuoles were observed in *Garin4* KO sperm heads.** The percentage of vacuole-containing sperm heads were analyzed. *Garin2-5b* (*Garin2*, *3*, *4*, *5a*, and *5b*) KO males were compared with WT males (One-way ANOVA, c, $P < 0.001$).
(TIF)

**S6 Fig. Amounts of globozoospermia-related protein, and the interaction between GARINs and CA/CN form of RAB2A/B.** (A) Expression of ADAM3 in mature spermatozoa was analyzed using cauda epididymal sperm lysates from WT and *Garin2-5b*$^{-/-}$ males. (B) Western blotting analysis was performed using testis lysates from WT and *Garin2-5b*$^{-/-}$ males. Globozoospermia-related proteins, GOPC, ZPBP1, and SPACA1 were blotted with their corresponding antibodies. Anti-ACTB was used to detect ACTB as an endogenous control. (C and D) Co-IP of GARIN1B, GARIN2, GARIN4, GARIN5A, and GARIN5B with constitutively active (CA)/constitutively negative (CN) forms of RAB2A (C) or RAB2B (D).
(TIF)

**S1 Table. Efficiency of generating *Garin2-5b* knockout mice.**
(XLSX)

**S2 Table. Proteins identified by whole sperm mass spectrometry.**
(XLSX)

**S3 Table. Information about primary antibodies used in this study.**
(XLSX)

**S4 Table. Sequences used for phylogenetic analyses.**
(XLSX)

**S5 Table. Information about guide RNAs and primers used in this study.**
(XLSX)

**S6 Table. Data of GO analysis.**
(XLSX)

## Acknowledgments

We thank Dr. Julio M. Castaneda for the critical reading of the manuscript. We also thank Akinori Ninomiya and Fuminori Sugihara for the mass spectrometry analysis (Core Instrumentation Facility, Research Institute for Microbial Diseases, Osaka University).

## Author Contributions

**Conceptualization:** Haoting Wang, Rie Iida-Norita, Daisuke Mashiko, Haruhiko Miyata, Masahito Ikawa.

**Formal analysis:** Haoting Wang, Rie Iida-Norita, Daisuke Mashiko, Anh Hoang Pham.

**Funding acquisition:** Haruhiko Miyata, Masahito Ikawa.

**Investigation:** Haoting Wang, Rie Iida-Norita, Daisuke Mashiko, Anh Hoang Pham.

**Methodology:** Haoting Wang, Rie Iida-Norita, Daisuke Mashiko, Haruhiko Miyata.

**Project administration:** Haruhiko Miyata, Masahito Ikawa.

**Resources:** Masahito Ikawa.

**Supervision:** Haruhiko Miyata, Masahito Ikawa.

**Visualization:** Haoting Wang.

**Writing – original draft:** Haoting Wang, Haruhiko Miyata, Masahito Ikawa.

**Writing – review & editing:** Haoting Wang, Rie Iida-Norita, Daisuke Mashiko, Anh Hoang Pham, Haruhiko Miyata, Masahito Ikawa.

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
