## [Decision Letter · Decision Letter 0]

23 Apr 2024

Dear Dr Miyata,

Thank you very much for submitting your Research Article entitled 'Golgi associated RAB2 interactor protein family contributes to murine male fertility to various extents by assuring correct morphogenesis of sperm heads' to PLOS Genetics.

The manuscript was fully evaluated at the editorial level and by four peer reviewers. As you will see, all four reviewers are positive, but note a number of concerns that should be straightforward to address in a revised manuscript that will be evaluated again by the reviewers.

We therefore ask you to modify the manuscript according to the review recommendations. Your revisions should address the specific points made by each reviewer.

Yours sincerely,

Gregory S. Barsh

Section Editor

PLOS Genetics

Gregory Copenhaver

Section Editor

PLOS Genetics

Reviewer's Responses to Questions

**Comments to the Authors:**

Reviewer #1: To the authors

This manuscript describes the functional roles of the mouse Garin gene family members with the testis-predominant expression: Garin2, Garin3, Garin4, Garin5a, and Garin5b. The authors pointed out that these genes are important for the correct morphogenesis of sperm heads and that some of the genes are distinctively involved in male fertility. Linked to the previous work conducted by the authors, experiments included in the present study are nicely designated, and the manuscript is clearly written. The reviewer can recommend this manuscript to be published in PLoS Genetics, if the authors appropriately reply to my comments/suggestions shown below.

Major points:

Figs. 1B-1I: In vitro models using tissue cultured cells are nicely working. However, the authors have any lines of information about the interactions between testis-predominant GARINs and RAB2A/B in the testis? Please show the data if the authors have or discuss about this issue.

Relevant to the above comment, the authors provided no data on localization of testis-predominant GARINs. The information would be important when we focus on the distinctive functions of each tested GARINs. Please show the data if the authors have or discuss about the GARINs’ localization.

Fig. 6B: How about the immunoblotting data of spermatozoa from other Garin KO strains? The comparison among the Garin2-5b KO mice would support the idea that Garin3 is functionally different from other testis-predominant Garins. Relevant to this issue, please discuss about why only particular ADAM proteins were affected by Garin3 deletion.

Minor points:

Page 2, line 26: Please consider to change “…for acrosome biogenesis and GARIN1B is…” to “…for acrosome biogenesis and that GARIN1B is…”.

Page 3, line 42: Please consider to change “2 proteins” to “two proteins”.

Reviewer #2: The present studies uncover the role of several members of the Golgi Associated RAB2 Interactors (GARINs) protein family in the development of sperm head morphogenesis during spermiogenesis and their role in mouse fertility. The authors generated mutant-null mice for GARIN2-5b and found that loss of function for each protein resulted in aberrant sperm head morphogenesis. Garin2 and Garin4 null males showed normal fertility, whereas Garin5a and Garin5b null mutant males were subfertile, and Garin3 null males were completely infertile. Further cellular and molecular analysis revealed abnormal acrosomal morphology in Garin3 knockout males, together with reduced ADAM proteins in Garin3 null sperm, which was consistent with defective sperm binding and zona penetration in the null sperm. In addition, vacuoles were observed in the sperm head of Garin4 knockout mice. The conclusions are supported by laborious transgenic efforts and indicate that GARINs play a critical role in sperm head morphogenesis and possibly gamete interaction. No major issues were found.

Minor comments:

-Fig 1E: What happened to the PA antibody staining in this immunoblot?

-Fig 2C: Right panel missing a scale bar.

-Fig 3A: Garin 3 null sperm appear to have issues in zona penetration. Did you find sperm stuck midway through the zona? Elaborate on this.

-CASA analyses might not be the best automated system to identify minor differences in sperm motility. If previous defects in motility were discovered using the same technology, some phrasing should be added in the discussion.

Reviewer #3: The authors have previously demonstrated the essential roles of GARIN1A and GARIN1B in acrosome formation in mouse spermatozoa. However, while other GARIN family proteins are present in mammalian sperm, their functions remain unreported. This paper investigates the function of these additional GARIN family proteins (GARIN2-5B) primarily using a gene targeting strategy, unveiling distinct phenotypes, including complete infertility in Garin3 KO male mice. Garin1b KO (previous work) and Garin3 KO mice both exhibiting infertility in males, affecting the expression of sperm membrane proteins. However, these proteins are specific to each case. Hence, it's hypothesized that each GARIN protein is involved in a unique process of acrosome or sperm head morphology. All experiments were meticulously conducted and merit publication in PLOS Genetics. Although further experiments are not required to perform, I recommend to add some more information to improve the paper.

Minor comments

The GRAIN family proteins are known by other gene names. In fact, in previous work by Morohoshi et al. (2021), the author used FAM71F1 and FAM71F2 instead of GARIN1B and GARIN1A, respectively. Similarly, in a recent study by Mo et al. (2023), FAM71D was used instead of GARIN2. While the authors mentioned Fam71 in the introduction section, this term was not utilized in either the title or abstract. I recommend adding the term 'FAM71' as an alternative name for GARIN in the abstract to facilitate the searching process.

Garin5a and Garin5b appear to be closely related paralogs. However, molecular phylogenetic analysis reveals significant differences in their amino acid sequences (Fig. S1). It would be beneficial if the author could provide some insight or comment on this observation.

Garin3 binds to both the active and inactive forms of RAB2 (Fig. S6). However, it's possible that the exact Kd value differs between the two forms. The authors should discuss the limitations of the experiments conducted in this study and suggest the type of experiment that can determine the difference in affinity between the two forms.

Reviewer #4: This manuscript contributes original and important insights into a protein family, the Golgi Associated RAB2 Interactors (GARINs), with several members that have a significant role in sperm head/acrosome structure during spermiogenesis and are necessary or important for fertility. This work builds upon and significantly advances previous studies and provides a detailed evaluation of the Garin 2-5b proteins in sperm physiology and fertility. The manuscript is well organized, the figures and tables are clear, and the writing is lucid. The methods are appropriate, rigorous, and thorough and the conclusions drawn are supported by the data.

There are no major issues to report but a few minor issues or comments:

1. Application of EFDs and PC to sperm head morphology is an effective quantitative approach to sperm head morphology, and while a previous use of it was referenced it would still be useful to provide some additional explanation on its application (e.g. scoring of parameters measured in Figure 4C).

2. PNA lectin is a very useful tool for evaluating acrosomal morphology and clearly demonstrates the differences in the Garin 2-5b knockouts. However, in the abnormal acrosomes, especially in Garin 3-/- sperm it would be useful to see a higher resolution evaluation of the acrosomal contents and membranes, perhaps by TEM. The PNA lectin is presumably labeling the outer acrosomal membrane, but since the expression of some acrosomal proteins has been affected in these knockouts, it is also possible that the carbohydrate moieties recognized by the lectin are also trafficked differently which may not correctly represent the actual shape of the acrosome. This is not essential for publication here but would be important to consider for future work.

3. The methodology used for the gene ontology (GO) analysis in Figure 6 is not explained and should be included in the methods section.

Overall, this is an excellent, comprehensive study of the GARIN protein family role in spermiogenesis and sperm function in fertility that merits publication in PLOS Genetics.

**Have all data underlying the figures and results presented in the manuscript been provided?**

Reviewer #1: Yes

Reviewer #2: Yes

Reviewer #3: Yes

Reviewer #4: Yes

PLOS authors have the option to publish the peer review history of their article (what does this mean?). If published, this will include your full peer review and any attached files.

Reviewer #1: No

Reviewer #2: **Yes: **Matteo Avella

Reviewer #3: No

Reviewer #4: No

---

## [Decision Letter · Decision Letter 1]

10 Jun 2024

Dear Dr Miyata,

We are pleased to inform you that your manuscript entitled "Golgi associated RAB2 interactor protein family contributes to murine male fertility to various extents by assuring correct morphogenesis of sperm heads" has been editorially accepted for publication in PLOS Genetics. Congratulations!

The revised manuscript was seen by the previous reviewers who are unanimous in their recommendations to move forward.

Yours sincerely,

Gregory Barsh

Section Editor

PLOS Genetics

Gregory Copenehaver

Section Editor

PLOS Genetics

Comments from the reviewers (if applicable):

Reviewer's Responses to Questions

**Comments to the Authors:**

Reviewer #1: The authos appropriately replied to all of my comments/suggestions. Therefore, this reviewer can reccomend this revised manuscript to be published in PLoS Genetics.

Reviewer #2: The Authors have addressed all the concerns raised by this reviewer.

Reviewer #3: The authors satisfactorily responded to all the requests and suggestions I made, as well as those from the other reviewers. Therefore, the revised manuscript is now ready for publication in PLOS Genetics.

Reviewer #4: The authors' responses to the review comments are satisfactory and this reviewer supports publication.

**Have all data underlying the figures and results presented in the manuscript been provided?**

Reviewer #1: Yes

Reviewer #2: Yes

Reviewer #3: Yes

Reviewer #4: Yes

PLOS authors have the option to publish the peer review history of their article (what does this mean?). If published, this will include your full peer review and any attached files.

Reviewer #1: No

Reviewer #2: No

Reviewer #3: No

Reviewer #4: No

**Data Deposition**

http://datadryad.org/submit?journalID=pgenetics&manu=PGENETICS-D-24-00145R1

**Press Queries**

---

## [Editor Report · Acceptance letter]

24 Jun 2024

PGENETICS-D-24-00145R1 

Golgi associated RAB2 interactor protein family contributes to murine male fertility to various extents by assuring correct morphogenesis of sperm heads 

Dear Dr Miyata, 

We are pleased to inform you that your manuscript entitled "Golgi associated RAB2 interactor protein family contributes to murine male fertility to various extents by assuring correct morphogenesis of sperm heads" has been formally accepted for publication in PLOS Genetics! Your manuscript is now with our production department and you will be notified of the publication date in due course.

With kind regards,

Zsofia Freund

PLOS Genetics

On behalf of:
